# Prostate Cancer Detection with mpMRI According to PI-RADS v2 Compared with Systematic MRI/TRUS-Fusion Biopsy: A Prospective Study

Anja Sauck [1,†], Isabelle Keller [1,†], Nicolin Hainc [2], Denis Pfofe [3], Arash Najafi [4], Hubert John [1,5,‡] and Joachim Hohmann [4,6,*,‡]

1   Clinic of Urology, Cantonal Hospital Winterthur, 8400 Winterthur, Switzerland
2   Department of Neuroradiology, Clinical Neuroscience Center, University Hospital Zurich, 8091 Zurich, Switzerland
3   Institute of Pathology, Cantonal Hospital Winterthur, 8400 Winterthur, Switzerland
4   Institute of Radiology and Nuclear Medicine, Cantonal Hospital Winterthur, 8400 Winterthur, Switzerland
5   Medical Faculty, University of Zurich, 8032 Zurich, Switzerland
6   Medical Faculty, University of Basel, 4056 Basel, Switzerland
*   Correspondence: joachim.hohmann@unibas.ch
†   These authors contributed equally to this work.
‡   Should be considered jointly as senior authors.

**Abstract:** Background: mpMRI assesses prostate lesions through their PI-RADS score. The primary goal of this prospective study was to demonstrate the correlation of PI-RADS v2 score and the volume of a lesion with the presence and clinical significance of prostate cancer (PCa). The secondary goal was to determine the extent of additionally PCa in inconspicuous areas. Methods: All 157 patients underwent a perineal MRI/TRUS-fusion prostate biopsy. Targeted biopsies as well as a systematic biopsy were performed. The presence of PCa in the probes was specified by the ISUP grading system. Results: In total, 258 lesions were biopsied. Of the PI-RADS 3 lesions, 24% were neoplastic. This was also true for 36.9% of the PI-RADS 4 lesions and for 59.5% of the PI-RADS 5 lesions. Correlation between ISUP grades and lesion volume was significant ($p < 0.01$). In the non-suspicious mpMRI areas carcinoma was revealed in 19.7% of the patients. Conclusions: The study shows that the PI-RADS v2 score and the lesion volume correlate with the presence and clinical significance of PCa. However, there are two major points to consider: First, there is a high number of false positive findings. Second, inconspicuous mpMRI areas revealed PCa.

**Keywords:** prostate cancer; mpMRI; PI-RADS; ISUP; MRI/TRUS-fusion biopsy

## 1. Introduction

Prostate cancer is the most common malignant tumor in western countries [1,2]. In 2011, approximately 900,000 new cases and 258,400 deaths due to prostate cancer were reported worldwide [3]. This number increased to 1,276,106 new cases resulting 358,989 deaths, accounting for 3.8% of all deaths caused by cancer in men, in 2018 [4]. Transrectal or perineal prostate biopsy is the gold standard for confirming the diagnosis in patients with elevated prostate specific antigen (PSA) and/or abnormal digital rectal examination (DRE) [5]. The use of multiparametric MRI (mpMRI) of the prostate has been established over the last 10 years for local staging of confirmed prostate cancer and screening/detection of prostate cancer. With improving image quality and new imaging sequences, the aim is not only to diagnose capsular penetration and infiltration of neighboring organs by prostate cancer with MRI but also to visualize the intra-prostatic size and localization of frequently occurring multifocal prostate cancer [6,7]. The European Society of Urogenital Radiology (ESUR) introduced the Prostate Imaging and Reporting Data System (PI-RADS) v1 in 2011 to standardize mpMRI findings, which resulted in a significant improvement

in tumor detection accuracy [7–9]. Consequently, this led to a higher acceptance rate of MRI among urologists as a viable imaging modality for the diagnosis of suspected prostate cancer [10–12].

A new version of the PI-RADS v1 classification, termed PI-RADS v2, was introduced in December 2014 as revised by the ESUR, the American College of Radiology (ACR) and the AdMeTech Foundation and was expected to achieve global standardization through international collaboration. Meanwhile PI-RADS v2.1 is the current version, with some slight changes to the definition of the PI-RADS scores compared to PI-RADS v2. However, we used the PI-RADS v2 version in this study, which was the most current version at the time of patient acquisition.

Besides the standardization of the lesion scores, advances in imaging technology have fostered the desire for targeted biopsies of lesions. This was initially restricted to cognitive fusion ultrasound or the time-intensive MRI-guided in-bore biopsies [13]. The development of software for the fusion of three-dimensional prostate models from MRI with the lesions superimposed onto the live image of transrectal ultrasound (TRUS) and the hardware for accurate biopsy via a template have revolutionized prostate biopsy procedures [14,15].

In this study, we investigate the correlation between the PI-RADS v2 categories and the volume of lesions, both determined by mpMRI, and the incidence and clinical significance of prostate cancer, both determined by the ISUP grades of the biopsies. The second aim of this study is to determine the extent of additionally found cancer in non-suspicious prostate areas.

## 2. Materials and Methods

The MRI/TRUS-fusion prostate biopsies were all performed by two urologists (A.S. and I.K.) using the BiopSee® system (MedCom GmbH, Darmstadt, Germany) and the BiopSee® ultrasound probe (MedCom GmbH, Darmstadt, Germany). The marking of the prostate and the suspicious lesions in the BiopSee® software was performed by one of the two urologists based on the findings described by the radiologists on the T2-weighted or DWI MRI sequences.

All patients received general or spinal anesthesia and were positioned in the lithotomy position for the biopsy procedure. Peri-interventional antibiotic prophylaxis with a single dose of 1.5 g cefuroxime was administered. The 3D-model of the T2-weighted MRI sequences was fused with the real-time TRUS. Each marked lesion was scheduled for two or more punch biopsies, depending on their size. The remainder of the prostate, which appeared inconspicuous on MRI, was biopsied systematically. Biopsies were taken via the perineal template and documented according to the BiopSee® system. Each punch biopsy was individually processed for histopathological assessment.

All biopsies were assessed according to the guidelines of the International Society of Urological Pathology (ISUP) Consensus Conference on Grading of Prostatic Carcinoma 2014 [16–18].

The patients age, prostate volume, recent PSA levels (at most three months before biopsy), DRE results, number of pre-biopsies in the past three years were recorded for all study participants (Table 1). For patients undergoing active surveillance, the number of positive core biopsies with the respective position and the ISUP grading were additionally documented.

During the intervention, the prostate volume and the number and volume of the individual lesions were recorded together with the PI-RADS v2 score. The positions and the histological diagnosis of the prostate lesions or systematic biopsies, as well as the radiological and the clinical T-stage were documented. Core biopsies positive for carcinoma obtained during the systematic biopsy taken less than 5 mm from a lesion were considered part of that lesion.

**Table 1.** Patients baseline characteristics.

| Variable | Value | Range or % |
|---|---|---|
| Men included in analysis, n, % | 157 | 100 |
| Age, years, median n, (IQR) | 65 | (58–70) |
| PSA level, ng/mL, median n, (IQR) | 7.07 | (4.91–10.4) |
| Suspicious DRE findings ($\geq$T2), n, % | 61 | 38.85 |
| Prostate volume, mL, median n, (IQR) | 43 | (31–64) |
| PSAD, ng/mL/mL, median n, (IQR) | 0.14 | (0.097–0.23) |
| Patients without prior biopsy, n, % | 118 | 75.16 |
| Patients with prior biopsy, n, % | 39 | 24.84 |
| Patients with 1 prior biopsy, n, % | 25 | 15.92 |
| Patients with 2 prior biopsies, n, % | 10 | 6.37 |
| Patients with 3 prior biopsies, n, % | 4 | 2.55 |
| Patients undergoing active surveillance, n, % | 9 | 5.73 |
| Days from mpMRI to biopsy, median n, (IQR) | 24 | (12–41) |
| mpMRI with no lesion, n, % | 11 | 7.01 |
| mpMRI with 1 lesion, n, % | 67 | 42.68 |
| mpMRI with 2 lesions, n, % | 55 | 35.03 |
| mpMRI with $\geq$3 lesions, n, % | 24 | 15.29 |
| Total number of lesions, n, % | 258 | 100 |
| PI-RADS v2 score 1, n, % | 1 | 0.39 |
| PI-RADS v2 score 2, n, % | 13 | 5.06 |
| PI-RADS v2 score 3, n, % | 50 | 19.4 |
| PI-RADS v2 score 4, n, % | 157 | 60.9 |
| PI-RADS v2 score 5, n, % | 37 | 14.3 |
| Total biopsies per patient, median n, (IQR) | 20 | (18–20) |
| Systematic biopsies per patient, median n, (IQR) | 11 | (8–13) |
| Biopsies per lesion, median n, (IQR) | 4 | (3–5) |

IQR, interquartile range; DRE, digital rectal examination; PSA, prostate specific antigen; mpMRI, multiparametric magnetic resonance imaging.

Patient data, mpMRI and histological biopsy results were analyzed descriptively by an independent external company specialized in statistical analysis (NOVUSTAT® Statistik-Beratung, Kreuzlingen, Switzerland). Correlation analysis was carried out using medcalc software (MedCalc Software Ltd., Ostend, Belgium) and R statistical software (R Core Team (2020). R: A language and environment for statistical computing. R Foundation for Statistical Computing, Vienna, Austria. https://www.r-project.org/, accessed on 14 August 2022) using the RStudio (R Studio Inc., Boston, MA, USA; https://www.rstudio.com/ https://www.r-project.org/, accessed on 14 August 2022) user interface for the Chi–square and the Fisher test. Discriminant analysis was used to calculate sensitivity and specificity. Contingencies between two groups were compared with a two-tailed Chi–square test. Data from multiple groups were compared using the Kruskal-Wallis test. A *p*-value of $\leq$0.05 was considered statistically significant.

## 3. Results

In this prospective study, 157 patients were ultimately enrolled and statistically evaluated out of a total of 228 patients, all of which underwent an MRI/TRUS-fusion prostate biopsy procedure between September 2015 and March 2017 (Figure 1; Table 1).

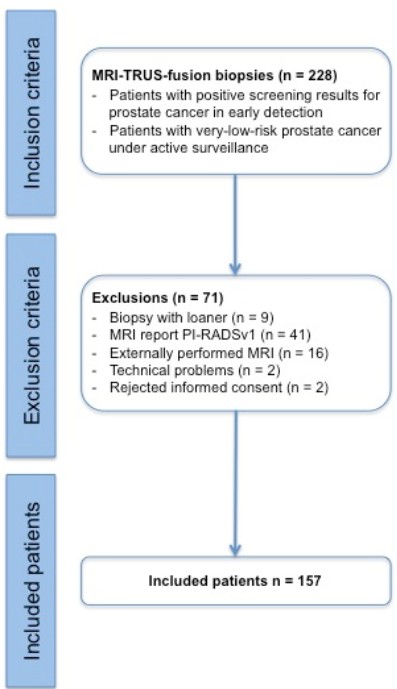

**Figure 1.** Modified CONSORT (Consolidated Standards of Reporting Trials) diagram indicating the number of participating patients and the reasons for study exclusions. Note: a "biopsy with loaner" is a biopsy with a replacement device and not with the device mentioned in Section 2.

Of the 157 patients, 118 (75.2%) had no previous biopsy. A total of 25/157 (15.9%) of the patients had 1 biopsy prior, 10/157 (6.4%) had 2 previous biopsies, and 4/157 (4.6%) had 3 or more previous biopsies. 9/157 (5.7%) patients were on active surveillance for known prostate cancer (Table 1). The median time between MRI and fusion biopsy was 24 (IQR 12–41) days.

In the 157 patients, a total of 258 lesions were detected across all PI-RADS scores. Of these 258 lesions, 14 lesions with a PI-RADS v2 score of 1 or 2 were reported (Table 2). These 14 lesions are not included in the calculations in Tables 3 and 4, since lesions with these scores are not considered suspicious for carcinoma. In 67/157 (42.7%) patients, one lesion was detectable on MRI, 55/157 (35.03%) had 2 lesions, and 24/157 (15.3%) had 3 or more lesions. In 11/157 (7%) patients, no lesions were detected on MRI.

**Table 2.** Tumor detection via PI-RADS v2 score and ISUP grading.

| mpMRI | | ISUP | | | | | |
|---|---|---|---|---|---|---|---|
| PI-RADS | n | 0 * | 1 | 2 | 3 | 4 | 5 |
| no lesion | 11 | 9 (81.8%) | 2 (18.2%) | - | - | - | - |
| 1 | 1 | 1 (100%) | - | - | - | - | - |
| 2 | 13 | 10 (76.9%) | 3 (23.1%) | - | - | - | - |
| 3 | 50 | 38 (76.0%) | 10 (20.0%) | 2 (4.0%) | - | - | - |
| 4 | 157 | 99 (63.1%) | 33 (21.0%) | 19 (12.1%) | 4 (2.5%) | 1 (0.6%) | 1 (0.6%) |
| 5 | 37 | 15 (40.5%) | 6 (16.2%) | 7 (18.9%) | 4 (10.8%) | 3 (8.1%) | 2 (5.4%) |
| total | | | | | | | |

* ISUP 0 = no carcinoma, benign/normal prostate parenchyma.

**Table 3.** Sensitivity and specificity for clinically significant prostate cancer (ISUP $\geq$ 2); mpMRI was suspect if PI-RADS v2 score $\geq$ 3, PSAD was suspect if $\geq$0.15 ng/mL/mL.

| Cancer-Predicting Variables | Sensitivity, % | Specificity, % | PPV, % | NPV, % |
|---|---|---|---|---|
| mpMRI | 95.5 | 15.5 | 45.7 | 82.4 |
| DRE | 49.3 | 70.0 | 65.0 | 55.0 |
| PSAD | 64.2 | 66.6 | 58.9 | 71.4 |
| DRE + PSAD | 32.8 | 92.2 | 75.9 | 64.8 |
| mpMRI + DRE | 49.3 | 72.2 | 56.9 | 65.7 |
| mpMRI + PSAD | 61.2 | 73.3 | 63.1 | 71.7 |
| mpMRI + DRE + PSAD | 34.3 | 50.0 | 77.4 | 13.2 |

DRE, digital rectal examination; PSAD, PSA density; PPV, positive predictive value; NPV, negative predictive value.

**Table 4.** Relationship between PI-RADS v2 scores of 3–5, lesion volume, ISUP grades and risk calculation for prostate cancer (PCa) in general and clinically significant prostate cancer (csPCa).

| PI-RADS Score | n | Lesion Volume, mL | 0 * | 1 | 2 | 3 | 4 | 5 | csPCa n | Total n | PCa Risk, % | csPCa Risk, % |
|---|---|---|---|---|---|---|---|---|---|---|---|---|
| | | | | | ISUP Grades | | | | | | | |
| 3 | 50 | ≤0.5 | 26 | 7 | 2 | 0 | 0 | 0 | 2 | 35 | 25.7 | 5.7 |
| | | 0.5–1 | 7 | 2 | 0 | 0 | 0 | 0 | 0 | 9 | 22.2 | 0.0 |
| | | ≥1 | 5 | 1 | 0 | 0 | 0 | 0 | 0 | 6 | 16.7 | 0.0 |
| 4 | 157 | ≤0.5 | 69 | 22 | 13 | 1 | 0 | 1 | 15 | 106 | 34.9 | 14.2 |
| | | 0.5–1 | 18 | 10 | 4 | 2 | 0 | 0 | 6 | 34 | 47.1 | 17.6 |
| | | ≥1 | 12 | 1 | 2 | 1 | 1 | 0 | 4 | 17 | 29.4 | 23.5 |
| 5 | 37 | ≤0.5 | 5 | 3 | 4 | 0 | 1 | 0 | 5 | 13 | 61.5 | 38.5 |
| | | 0.5–1 | 3 | 3 | 0 | 0 | 0 | 0 | 0 | 6 | 50.0 | 0.0 |
| | | ≥1 | 7 | 0 | 3 | 4 | 2 | 2 | 11 | 18 | 61.1 | 61.1 |
| total | 244 | ≤0.5 | 100 | 32 | 19 | 1 | 1 | 1 | 22 | 154 | 35.1 | 14.3 |
| | | 0.5–1 | 28 | 15 | 4 | 2 | 0 | 0 | 6 | 49 | 42.9 | 12.2 |
| | | ≥1 | 24 | 2 | 5 | 5 | 3 | 2 | 15 | 41 | 41.5 | 36.6 |

* ISUP 0 = no carcinoma, benign/normal prostate parenchyma.

*3.1. MRI Lesions Reported with*

3.1.1. PI-RADS v2 Score 5

A carcinoma was diagnosed histologically in 22/37 (59.5%) lesions with a PI-RADS v2 Score of 5. In 3/37 (8.1%) lesions an atypical small acinar proliferation (ASAP) and in a further 3/37 (8.1%) high grade prostatic intraepithelial neoplasia (high grade PIN) was identified in addition to, or without, carcinoma detection. Additionally, in 8/37 (21.6%) lesions an active and/or chronic inflammation was diagnosed independently of other neoplastic changes.

3.1.2. PI-RADS v2 Score 4

Out of 157 lesions with a PI-RADS v2 score of 4, 58/157 (36.9%) showed evidence of carcinoma with a relatively high rate (63.1%) of benign histology (Table 2). A total of 27/157 (17.2%) biopsies showed ASAP and/or high-grade PIN, and 23/157 (14.6%) had active and/or chronic inflammation.

3.1.3. PI-RADS v2 Score 3

50 lesions had a PI-RADS v2 score of 3. In these lesions a carcinoma was detected in 12/50 (24%) of cases. At the same time or separately in 4/50 (8%) lesions the presence of ASAP and/or high-grade PIN respectively was diagnosed. Additionally, in 7/50 (14%) lesions active and/or chronic inflammation could be detected.

### 3.1.4. PI-RADS v2 Score 2

In total, only 13 lesions had a PI-RADS v2 score of 2. In 3/13 (23.1%) lesions a carcinoma could be detected histologically, whereas in the remaining 10/13 (79.9%) carcinoma was ruled out. ASAP was detected in 1/13 (7.7%) of the lesions, but no high-grade PIN could be documented. In 4/13 (30.8%) of the lesions active and/or chronic inflammation could be detected.

### 3.2. Inconspicuous mpMRI Examinations and Biopsies Outside of Lesions

A systematic biopsy demonstrated ISUP grade 1 carcinoma in 2/11 (18.2%) patients with negative mpMRI of the prostate (no PI-RADS v2 lesion) and in 1/11 (9.1%) patients a clinically relevant carcinoma with tumor stage cT2c was found.

It should also be mentioned that systematic biopsy outside of the lesions demonstrated additional regions of the prostate affected by carcinoma in 31/157 patients (19.7%) and thus lead to a higher tumor stage. The systematic biopsy is crucial in these cases as it affects further management. In our cohort 8/157, (5%) patients demonstrated clinically relevant prostate cancer on systematic biopsy requiring treatment, whereas their PI-RADS v2 lesion biopsies were benign.

### 3.3. Sensitivity and Specificity of mpMRI for Prostate Cancer Detection

In our study, the mpMRI with lesions PI-RADS v2 scores of 3, 4 or 5 has a high sensitivity for carcinoma detection of 95.5%, but a specificity of just 15.5%. In case of an inconspicuous mpMRI (no lesion or lesions with a PI-RADS v2 score of 1 or 2), a significant prostate carcinoma (csPCa), which means an ISUP grade $\geq$ 2, can be excluded with 82.4% certainty (NPV, Table 3). In comparison, the digital rectal examination alone has a sensitivity of 49.3% and a specificity of 70.0%, whereas the PSAD alone with a cut-off value of 0.15 ng/mL/mL (pathological $\geq$ 0.15 ng/mL/mL) achieved 64.2% and 66.6%, respectively.

### 3.4. MRI Lesion Volume Relation with ISUP Grades and Cancer Prediction

We observed a positive relationship between increasing lesion size and the occurrence of higher ISUP gradings (Figure 2).

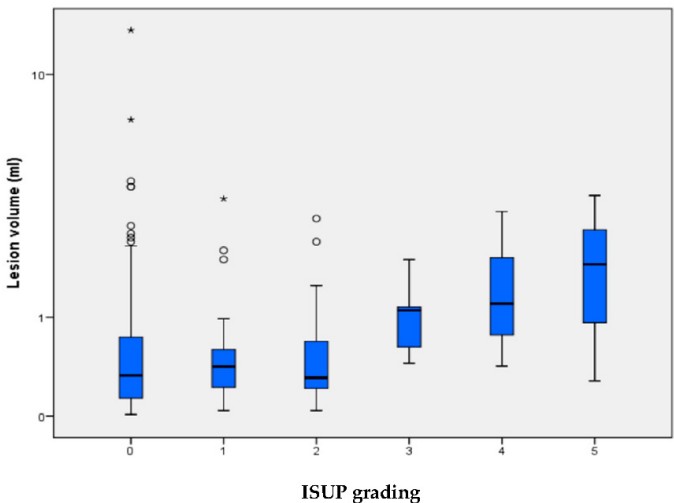

**Figure 2.** Lesion volume is indicated for each of the ISUP grading groups, for which ISUP 0 signifies no carcinoma/benign histology. The median ($\pm$IQR) lesion volume with whiskers within 1.5 times the IQR. Circles are lesions between 1.5 and 3 times, asterisks are lesions with >3 times IQR deviation from the third quartile. One benign outlier with a lesion volume with 42 mL is not depicted in the graphical representation for clearer visualization. The lesion volume correlates with ISUP grading ($p$ = 0.0076, Kruskal-Wallis-Test).

A significantly higher rate of carcinoma detection could be determined by the univariate regression analysis with an MRI lesion size $\geq$ 1 mL ($p$ = 0.0012) within suspicious MRI lesions with a PI-RADS v2 score $\geq$ 3 (Table 4). When only MRI lesions $\geq$ 1 mL were considered, the detection rate of clinically relevant prostate cancer in lesions with PI-RADS v2 score of 5 increased from 43.2% to 61.1%, and in lesions with PI-RADS v2 score of 4 from 15.9% to 23.5%. In line with these findings, we found a significant association between lesion volume > 1 mL and the presence of clinically relevant carcinomas ($p$ = 0.0247, Chi-square test) for lesions with a PI-RADS v2 score $\geq$ 4. Irrespective of the PI-RADS v2 score, the relative risk of a clinically relevant prostate carcinoma for lesions $\geq$ 1 mL compared to lesions < 1 mL is 1.381.

## 4. Discussion

In our prospective study comparing PI-RADS v2 scores with biopsy results, we found that the likelihood of finding clinically relevant prostate cancer correlates with the PI-RADS v2 score. Carcinoma was found in 59.5% of PI-RADS 5 lesions, 36.9% of PI-RADS 4 lesions, and 24% of PI-RADS 3 lesions. Additionally, we found a significant association between lesion volume > 1mL and the presence of clinically relevant carcinomas for lesions with a PI-RADS v2 score $\geq$ 4. These findings are in agreement with the previously reported studies of other cohorts [19–22].

mpMRI is a technological advancement over the standard MRI. Previous studies have shown that mpMRI can be used for the diagnosis and detection of prostate cancer [19,23]. mpMRI has an enhanced diagnostic efficiency, with a better detection of prostate cancer compared to ultrasound [22,24–26] and a proven correlation between the PI-RADS v2 score and the discovery of a prostate carcinoma [7,27,28]. A strongly but not significantly correlation between PI-RADS score and tumor aggressiveness, determined by higher ISUP grades, could be found in our study, as also reported in previous studies [20,21]. It has been shown that MRI/TRUS-fusion biopsies have a higher detection rate of clinically significant prostate cancer than normal ultrasound-guided prostate biopsy [23,29–31].

In our cohort, prostate cancer was found in 18% (2/11) of patients with a completely inconspicuous mpMRI (no lesions), however, these cancers were all ISUP 1. In 9% (1/11) the carcinoma was found on both sides. In addition to the targeted lesion biopsy, patients with lesions on the mpMRI underwent a systematic biopsy procedure explicitly focused on areas outside of the lesion. The systematic biopsy results led to a higher tumor stage in 20% (31/157) of the patients. In 5% (8/157) of patients, mpMRI lesion biopsies were carcinoma-free, but systematic biopsies outside of these lesions were positive for prostate carcinoma. These results are similar to those of Sonn et al. [29], Miyagawa et al. [27], and Kuru et al. [28]. However, in the latter study, only patients with inconspicuous mpMRI were subjected to a systematic biopsy. Based on our data, we recommend that all patients who are selected for fusion biopsy due to a suspicious lesion also undergo a systematic biopsy.

The decision to carry out a prostate biopsy with inconspicuous mpMRI should be made on an individual basis in consultation with the patient while considering predictive factors such as family history, DRE and PSAD. In the inconspicuous mpMRI or mildly suspicious PI-RADS v2 scores of 1 and 2, we detected only ISUP 1 prostate cancers. This was, however, found bilaterally in some cases, which theoretically means an upgrading of the tumor stage from a stage T2a to a stage T2c but does not change the therapy concept of active surveillance.

Carcinoma was detected in only 12/50 lesions (24%) with a PI-RADS v2 score 3. These were carcinomas of ISUP 1 (20%) and ISUP 2 (4%). Liddell et al. concluded from the application of the Epstein criteria in a retrospective cohort that a PI-RADS v2 score 3 lesion does not require biopsy and that these lesions should be monitored regularly due to the low risk of clinically significant prostate cancer (20). However, whether the Epstein criteria are suitable for the identification of clinically insignificant prostate cancer has been questioned in several studies [30–32].

In addition, we could show in this study that the ISUP grading of prostate cancer increases with the lesion size as measured with mpMRI. Carcinoma detection in suspicious MRI lesions with a PI-RADS v2 score $\geq$ 3 is significantly higher for lesions with a volume $\geq$ 1 mL compared to smaller lesions ($p$ = 0.0012). With a PI-RADS v2 score of 3 and a lesion volume $\leq$ 0.5 mL, the risk of significant prostate cancer is 5.7% (Table 4). With a PI-RADS v2 core of 5 and a lesion size $\geq$ 1 mL, there was a significantly higher risk ($p$ = 0.035) for the presence of a clinically relevant prostate carcinoma with 61.1%. Thus, the occurrence of a clinically relevant carcinoma in a PI-RADS v2 score 5 with a lesion size $\geq$ 1 mL is significantly more likely than with a lower PI-RADS v2 score and the same lesion size.

In our study, mpMRI has a sensitivity of 95.5% for the presence of prostate cancer (Table 4). Thus, almost all prostate cancers are detected on mpMRI. However, the positive predictive value of mpMRI is less than 50%, which is a clear weakness of the PI-RADS v2 diagnosis. In contrast, the negative predictive value is much better with a value beyond 80%, but not optimal. The question remains concerning whether a biopsy should be performed after inconspicuous mpMRI or only based on PI-RADS v2 score > 3, as shown by Liddell et al. [21]. In the study by Sonn et al. [33] the authors found that the detection of clinically relevant prostate cancer is not only dependent on the PI-RADS v2 score, but also on the duration of active surveillance and the radiologist. Indeed, Sonn et al. showed that different radiologists evaluated identical lesions in the mpMRI with different PI-RADS v2 scores. This resulted in, for example, tumor detection rates ranging from 40 to 80% based on a PI-RADS v2 score of 5. Note that in study by Sonn et al., PI-RADS v1 and PI-RADS v2 findings were performed (from the introduction of PI-RADS v2 in 2015). The previously performed mpMRI were not retrospectively reviewed with PI-RADS v2.

The PI-RADS v2 criteria have since been revised to PI-RADS v2.1 to facilitate the recognition of healthy patients and thus reduce the number of unnecessary prostate biopsies. The value of assessing PSAD and lesion size could be shown in our study and could prove to be useful to this end.

The problem of interreader variability with regards to the PI-RADS criteria and thus different PI-RADS v2 scores among radiologists will likely not be solved by applying new criteria. However, a study by Radtke et al. in 2017 [34] showed that the use of new risk models including clinical parameters, together with an mpMRI, may lead to a reduction in diagnostic biopsies. Diagnostic software programs that support and facilitate the interpretation of a prostate MRI have entered the market recently [35]. It remains to be seen if these programs can achieve increased accuracy and specificity. Thus, we believe that this should be reviewed in future studies.

One of the weaknesses of our study is that it is a single-center study. Furthermore, our study investigated a pre-selected heterogenous patient cohort based on a referral from a general practitioner, positive family history, suspicious lab results or even patients undergoing active surveillance for previously diagnosed prostate cancer. In addition, some patients were biopsy naive, and some had repeated biopsies. The mpMRI were assessed by different radiologists, which is often the reality in many clinics. Our evaluations of unremarkable mpMRI, as well as mpMRI with PI-RADS v2 score 1 or 2 should be considered critically as we report only a very small number of cases.

## 5. Conclusions

The probability of having prostate cancer increases with the level of PI-RADS v2 score and lesion size. The negative predictive value of mpMRI is better (82.4%) than the positive predictive value (45.7%). However, additional prostate cancer was detected in non-suspicious areas in 20% of patients. The use of mpMRI with the PI-RADS v2 still has its weaknesses. In particular, we believe efforts should be focused on better detection of carcinoma-free prostates with this approach, which is already carried out by the introduction of PI-RADS v2.1. Further evaluations with the newly defined and more or less slightly changed definition of PI-RADS scores will show if this improves the situation.

**Author Contributions:** Conceptualization, A.S., I.K., H.J. and J.H.; methodology, A.S, I.K., N.H., D.P. and J.H.; validation, A.S., A.N., D.P., I.K. and J.H.; formal analysis, A.S., A.N., D.P. and J.H.; investigation, A.S., I.K. and D.P.; resources, A.S., I.K. and H.J.; data curation, N.H., A.N., D.P. and J.H.; writing—original draft preparation, A.S., N.H. and J.H.; writing—review and editing, all authors; supervision, H.J. and J.H.; project administration, A.S. and J.H. All authors have read and agreed to the published version of the manuscript.

**Funding:** This research received no external funding.

**Institutional Review Board Statement:** The study was conducted in accordance with the Declaration of Helsinki and approved by Swissethics (Cantonal Ethics Committee of Zurich, No. BASEC 2016-01098) on 27 September 2016.

**Informed Consent Statement:** Informed consent was obtained from all subjects involved in the study.

**Data Availability Statement:** The datasets used and/or analyzed during the current study are available from the corresponding author on reasonable request.

**Conflicts of Interest:** The authors declare no conflict of interest.

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
