# Peer review of "Prostate Cancer Detection with mpMRI According to PI-RADS v2 Compared with Systematic MRI/TRUS-Fusion Biopsy: A Prospective Study"

_tomography, doi:10.3390/tomography8040169_

Round 1

Reviewer 1 Report

Manuscript ‘Prostate cancer detection with mpMRI according to PI-RADS v2 compared with systematic MRI/TRUS-fusion biopsy: a prospective study’

Sauck et al.

This paper describes the findings of a prospective mpMRI study in 157 patients from a single center in Switzerland. Their main finding is a positive relationship between PI-RADS v2 score as well as the volume of a lesion and the presence and severity of prostate cancer. In addition, a high number of false positives and false negatives is reported, the latter with respect to the finding of prostate cancer in inconspicuous mpMRI areas. Although the results of the study are in line with the current literature and not very novel, its prospective nature may make it suitable for publication in the special issue of Tomography: ‘Tumor diagnosis and treatment: imaging assessment’.

Minor:

1)    Methods.Table 1 can probably be omitted. This should be common knowledge for the interested reader.

2)    Results. Lines 119-121. The reference to figure 1 should be added here.

Line 163. Bland mpMRI is a negative mpMRI?

3)    Discussion. Lines 237-239. How does this statement relate to the findings of table 4?

The authors could elaborate a little more on the problem of false positives and negatives. Maybe PSMA PET/MR or an additional PSMA PET/CT could be of help?

Author Response

Dear Reviewer,

first, thank you for your time and effort in reviewing our manuscript. Thank you also for your helpful comments and suggestions for improvement of our work.

1) Unless it is really necessary, we would like to keep table 1, as most readers who are familiar with the subject will certainly know the meaning of the ISUP graduations, but the exact definition is probably no longer familiar to most.
2) Thank you for pointing this out. We have referenced figure 1 in the place mentioned, and also table 2, as the patients are also listed here. We hope that this is also in your sense.
3) Table 4 shows that a negative DRE and a low PSAD result in a high specificity, but unfortunately only a low sensitivity. A “positive” mpMRI with PI-RADS 4 or 5 lesions has a high sensitivity but only a low specificity. Accordingly, a prostate biopsy should be performed, and this is the classic approach, if the DRE is not normal, the PSA or PSAD is high or the mpMRI is positive. Other factors, such as a positive family history, are not mentioned in Table 4, but should also be considered in the individual decision for a prostate biopsy.

Thank you for the suggestion to compare mpMRI with PSMA PET/MR/CT in terms of false positive and false negative findings. However, we explicitly did not want to compare the initial methods for the detection of PCa with those for complete staging such as PSMA PET/MR/CT. On the one hand, we were fearing that this would have diluted our results, and on the other hand, we think that it would have dragged out the discussion section too much.

Reviewer 2 Report

In this study the authors evaluated the accuracy of PCa detection across different PIRAD scores. Although the study is not novel and the topic has extensively evaluated in the literature,  such study is still relevant to have an idea of mpMRI performance in different settings and different institutions. 

Some minor comments need to be noted:

1-    In line 21, the term “severity of prostate cancer” is not commonly used as there is no such thing as “severe prostate cancer. Terms such as aggressive or clinically significant are more appropriate

2-    In Figure 1, exclusion section. The term “biopsy with loaner” is not clear, what is biopsy with loaner

3-    Line 104, “biopsy T stage” should be replaced with “clinical T stage”

4-    The authors should mention in the limitations the inclusion of heterogenous patient cohort, such as Biopsy naïve, repeated biopsy, patient on active survaillance

5-    Line 147, the authors mentioned ISUP 0, which does not exist

Author Response

Dear Reviewer,

first, thank you for your time and effort in reviewing our manuscript. Thank you also for your helpful comments and suggestions for improvement of our work.

1) Thank you for this comment, you are quite right. We have replaced "severity" with "clinical significance" in the three places where this occurs.
2) Thank you for pointing this out. Apparently "loaner" is not a widely used word, yet it captures the essence of the matter best. We have explained "loaner" in more detail in the caption.
3) This also seems to us to be the better term, thank you very much. We have changed this.
4) We already mentioned the heterogeneous patient group. However, we tried to make that clearer.
5) You are right, ISUP 0 does not exist, we named the benign/normal parenchyma findings that way for evaluation purposes, but only explained it in Table 3 and Figure 2. It is poorly placed and rather difficult to understand in the place mentioned. We have used "benign histology" instead.